

# On tuning of atmospheric inverse methods: Comparison on ETEX and Chernobyl datasets using FLEXPART v8.1 and v10.3

Ondřej Tichý[1], Lukáš Ulrych[1], Václav Šmídl[1], Nikolaos Evangeliou[2], and Andreas Stohl[3]

[1]Institute of Information Theory and Automation, Czech Academy of Sciences, Prague, Czech Republic
[2]NILU: Norwegian Institute for Air Research, Kjeller, Norway
[3]Department of Meteorology and Geophysics, University of Vienna, Vienna, Austria

**Correspondence:** Ondřej Tichý (otichy@utia.cas.cz)

**Abstract.** Estimation of the temporal profile of an atmospheric release, also called the source term, is an important problem in environmental sciences. The problem can be formalized as a linear inverse problem where the unknown source term is optimized to minimize the difference between the measurements and the corresponding model predictions. The problem is typically ill-posed due to low sensor coverage of a release and due to uncertainties e.g. in measurements or atmospheric transport

modeling, hence, all state-of-the-art methods are based on some form of regularization of the problem using additional information. We consider two kinds of additional information, the prior source term, also known as the first guess, and regularization parameters for shape of the source term. While the first guess is based on information independent of the measurements, such as physics of the potential release or previous estimations, the regularization parameters are often selected by the designers of the optimization procedure. In this paper, we provide a sensitivity study of two inverse methodologies on the choice of the prior

source term as well as regularization parameters of the methods. The sensitivity is studied in two cases: data from the European Tracer Experiment (ETEX) using FLEXPART v8.1 and the caesium-134 and caesium-137 dataset from the Chernobyl accident using FLEXPART v10.3.

## 1 Introduction

The source-term describes the spatio-temporal distribution of an atmospheric release, and it is of great interest in the case of an

accidental atmospheric release. The aim of inverse modeling is to reconstruct the source term by maximization of agreement between the ambient measurements and prediction of an atmospheric transport model in a so called top–down approach (Nisbet and Weiss, 2010). Since information provided by the measurements is often insufficient in both spatial and temporal domains (Mekhaimr and Wahab, 2019), additional information and regularization of the problem is crucial for reasonable estimation of the source term (Seibert et al., 2011). Otherwise, the top-down determination of the source term can produce artifacts, often

resulting in some completely implausible values of the source term. One of the common regularization is the knowledge of the prior source term, also known as the first guess, considered within the optimization procedure (Eckhardt et al., 2008; Liu et al., 2017; Chai et al., 2018). However, this knowledge could dominate the resulting estimate and even outweigh the information present in the measured data. The aim of this study is to discuss drawbacks which may arise in setting of the prior source term





and to study the sensitivity of inversion methods on the choice of the prior source term. We utilize the ETEX (European Tracer Experiment) and Chernobyl datasets for demonstration.

We assume that the measurements can be explained by a linear model using the concept of the source-receptor-sensitivity (SRS) matrix calculated from an atmospheric transport model (e.g., (Seibert and Frank, 2004)). The problem can be approached

by constrained optimization with selected penalization term on the source term (Davoine and Bocquet, 2007; Ray et al., 2015; Henne et al., 2016) and further with an additional smoothness constraint (Eckhardt et al., 2008) used for both spatial (Stohl et al., 2011) and temporal (Seibert et al., 2011; Stohl et al., 2012; Evangeliou et al., 2017) profile smoothing. The optimization terms are typically weighted by covariance matrices whose forms and estimation have been studied in literature. Diagonal covariance matrices have been considered by Michalak et al. (2005) and its entries estimated using maximum likelihood method.

Since the estimation of full covariance matrices tends to diverge (Berchet et al., 2013), approaches using a fixed common autocorrelation timescale parameter for non-diagonal entries has been introduced (Ganesan et al., 2014; Henne et al., 2016) for atmospheric gas inversion. Uncertainties can be also reduced with the use of ensemble techniques, see, e.g., (Evensen, 2018; Carrassi et al., 2018) and references therein, when such an ensemble is available in the form of several meteorological input data sets and/or variations in atmospheric model parameters. Even with only one SRS matrix, the problem can be formulated as

a probabilistic hierarchical model with unknown parameters estimated together with the source term with constraints such as positivity, sparsity or smoothness (Tichý et al., 2016). The drawback of these methods is the necessity of selection and tuning of various model parameters, with the selection of the prior source term and its uncertainty being one of the most important.

There are various assumptions on the level of knowledge on the prior source term used in the inversion procedure. While in some papers no prior source term is assumed, this can be interpreted as the assumption of the zero prior source term (Bocquet,

2007; Tichý et al., 2016) with preference of zero solution on elements where no sufficient information from data are available. This assumption is typically formalized as the Tikhonov (Golub et al., 1999) or LASSO (Tibshirani, 1996) regularizations or their variants. Soft assumptions in the form of scale of the prior source term (Davoine and Bocquet, 2007), bounds on emissions (Miller et al., 2014), or even knowledge of total released amount as discussed e.g. in (Bocquet, 2005) can be considered. One can also assume the ratios between species in multi–species source term scenarios (Saunier et al., 2013; Tichý et al., 2018).

However, the majority of inversion methods assumes the knowledge of the prior source term explicitly (Connor et al., 2008; Eckhardt et al., 2008; Liu et al., 2017). This is more or less justified by appropriate construction of the prior source term, based, for example, on a detailed analysis of an inventory and accident (Stohl et al., 2012), on previous estimates when available (Evangeliou et al., 2017), or on measured or observed data (Stohl et al., 2011). While in well documented cases this approach could be well justified, in cases with very limited available information or even complete absence of information on the source

term such as the iodine occurrence over Europe in 2017 (Masson et al., 2018) or unexpected detection of ruthenium in Europe in 2017 (Bossew et al., 2019; Saunier et al., 2019), the use of strong prior source term assumptions could lead to prior dominated results with limited validity. Although the choice of the prior source term is crucial, few studies have discussed the choice of the prior source term in detail and provided sensitivity studies on this selection as in (Seibert et al., 2011) for the temporal profile of sulfur dioxide emissions or in (Stohl et al., 2009) for spatial distribution of greenhouse gas emissions.





The aim of this paper is to explore the sensitivity of linear inversion methods to the prior source term selection coupled with tuning of the covariance matrix representing modeling error. We considered the optimization method proposed by Eckhardt et al. (2008), its probabilistic counterpart formulated as the hierarchical Bayesian model, extended here by non–zero prior source term, with variational Bayes inference (Tichý et al., 2016) and with Monte Carlo inference using Gibbs sampler (Ulrych

and Šmídl, 2017). Two real cases will be examined: ETEX (Nodop et al., 1998) and Chernobyl (Evangeliou et al., 2016) datasets. The ETEX experiment provides ideal data for a prior source term sensitivity study since the emission profile is exactly known. We propose various modifications of the known prior source term and study their influence on the results of the selected inversion methods. The Chernobyl dataset provides, on the other hand, a very demanding case where only consensus on the release is available and source term is more speculative.

## 2   Inverse modeling using prior source term

We are concerned with linear models of atmospheric dispersion using a SRS matrix (Seibert, 2001; Wotawa et al., 2003; Seibert and Frank, 2004) which has been used in inverse modeling (Evangeliou et al., 2017; Liu et al., 2017). Here, the atmospheric transport model calculates the linear relationship between potential sources and atmospheric concentrations. The source-receptor sensitivity is calculated as $m_{ij} = c_i/x_j$ where $x_j$ is the assumed release from the release site at time $j$ and $c_i$

is the calculated concentration at a receptor $c_i$ at respective time period. The measurement $y_i$ at given time and location can be explained as a sum of contributions from all elements of the source term weighted by $m_{ij}$. In matrix notation

$$\mathbf{y} = \mathbf{M}\mathbf{x} + \mathbf{e}, \tag{1}$$

where $\mathbf{y} \in \Re^p$ is a vector aggregating measurements from all locations and times (in arbitrary order), $\mathbf{x} \in \Re^n$ is a vector of all possible releases from given time period, and all possible source-receptor sensitivities form the SRS matrix $\mathbf{M} \in \Re^{p \times n}$.

The residual model, $\mathbf{e} \in \Re^p$, is a sum of model and measurement errors. While the model looks trivial, its use in practical applications poses significant challenges. The key reason is that all elements of the model are subject to uncertainty (Winiarek et al., 2012; Liu et al., 2017) and the problem is ill-posed.

In the rest of this Section, we will discuss an influence of the modeling error and show how existing methods approach compensation of such error. We will analyze in detail two methods for the source term estimation: (i) the optimization model

proposed by Eckhardt et al. (2008) with a prior source term already considered and (ii) a Bayesian model (Tichý et al., 2016) extended here by a prior source term information and solved using both the variational Bayes method or the Gibbs sampling method.

### 2.1   Influence of atmospheric model error

It is generally assumed that the SRS matrix $\mathbf{M}$ is correct and the true source term minimizes error of $\mathbf{y} = \mathbf{M}\mathbf{x}$. However, $\mathbf{M}$ is

prone to errors due to a number of approximations in the formulation of the atmospheric transport model and use of uncertain





weather analysis data as input to the atmospheric transport model. Therefore, one should rather consider a hypothetical model

$$\mathbf{y} = (\mathbf{M} + \Delta_{\mathbf{M}})\,\mathbf{x}, \tag{2}$$

where $\mathbf{M}$ is the available estimate of the sensitivity matrix from a numerical model, and the term $\Delta_{\mathbf{M}}$ is the deviation of the estimate from the true generating matrix, $\mathbf{M}_{\text{true}} = (\mathbf{M} + \Delta_{\mathbf{M}})$. Exact estimation of $\Delta_{\mathbf{M}}$ is not possible due to lack of data,

however many existing regularization techniques can be interpreted as various simplified parametrizations of $\Delta_{\mathbf{M}}$.

The L2 norm[1] of the residuum between measurement and reconstruction for (3) would become

$$||\mathbf{y} - \mathbf{M}\mathbf{x} - \Delta_{\mathbf{M}}\mathbf{x}||_2^2 = ||\mathbf{y} - \mathbf{M}\mathbf{x}||_2^2 - 2\mathbf{y}^T\Delta_{\mathbf{M}}\mathbf{x} + \mathbf{x}^T\Xi\mathbf{x}$$
$$\Xi = \mathbf{M}^T\Delta_{\mathbf{M}} + \Delta_{\mathbf{M}}^T\mathbf{M} + \Delta_{\mathbf{M}}^T\Delta_{\mathbf{M}}. \tag{3}$$

The ideal optimization problem (right hand side of (3)) can be decomposed into the norm of residues of the estimated model

$||\mathbf{y} - \mathbf{M}\mathbf{x}||_2^2$, term linear in $\mathbf{x}$ (i.e., $-2\mathbf{y}^T\Delta_{\mathbf{M}}\mathbf{x}$) and term quadratic in $\mathbf{x}$ (i.e.,$\mathbf{x}^T\Xi\mathbf{x}$). Both of the additional terms contribute to incorrect estimation of $\mathbf{x}$ when $\Delta_{\mathbf{M}}$ is significant.

An common attempt to minimize the influence of the linear term is to define the prior source term $\mathbf{x}^a$, and subtract $\mathbf{M}\mathbf{x}^a$ from both sides of (2):

$$\mathbf{y} - \mathbf{M}\mathbf{x}^a = \mathbf{M}\,(\mathbf{x} - \mathbf{x}^a) + \Delta_{\mathbf{M}}\mathbf{x}. \tag{4}$$

Using substitutions $\overline{\mathbf{x}} = \mathbf{x} - \mathbf{x}^a$ and $\overline{\mathbf{y}} = \mathbf{y} - \mathbf{M}\mathbf{x}^a$,

$$\overline{\mathbf{y}} = \mathbf{M}\overline{\mathbf{x}} + \Delta_{\mathbf{M}}\,(\overline{\mathbf{x}} + \mathbf{x}^a)$$

yielding a new decomposition of (3):

$$||\mathbf{y} - \mathbf{M}\mathbf{x} - \Delta_{\mathbf{M}}\mathbf{x}||_2^2 = ||\mathbf{y} - \mathbf{M}\mathbf{x}||_2^2 - 2\mathbf{y}^T\Delta_{\mathbf{M}}(\overline{\mathbf{x}} - \mathbf{x}^a) +$$

$$+ (\overline{\mathbf{x}} - \mathbf{x}^a)^T\Delta_{\mathbf{M}}^T\Delta_{\mathbf{M}}(\overline{\mathbf{x}} - \mathbf{x}^a)^T. \tag{5}$$

Terms independent of $\mathbf{x}$ do not cause estimation error, and we can neglect them. The decomposition is then

$$||\mathbf{y} - \mathbf{M}\mathbf{x} - \Delta_{\mathbf{M}}\mathbf{x}||_2^2 = ||\mathbf{y} - \mathbf{M}\mathbf{x}||_2^2 - 2\tilde{\mathbf{y}}\Delta_{\mathbf{M}}\overline{\mathbf{x}} + \overline{\mathbf{x}}^T\Xi\overline{\mathbf{x}},$$
$$\tilde{\mathbf{y}} = (\mathbf{y}^T + \mathbf{x}^a\Delta_{\mathbf{M}}^T), \quad \Xi = \Delta_{\mathbf{M}}^T\Delta_{\mathbf{M}}, \tag{6}$$

which implies a different balance between the linear and quadratic terms than in the previous case.

In the ideal situation, we would like to optimize the left hand side of Eq. (3) and (5). However, due to unavailability of $\Delta_{\mathbf{M}}$ the linear term is typically ignored (assumed to be negligible) and the quadratic term is approximated using a parametric form of $\Xi$. The optimization criterion is then:

$$J = ||\overline{\mathbf{y}} - \mathbf{M}\overline{\mathbf{x}}||_2^2 - \overline{\mathbf{x}}^T\Xi\overline{\mathbf{x}}. \tag{7}$$

---
[1]The analysis can be generalized to a quadratic norm with arbitrary kernel $\mathbf{R}$, however, we will discuss its simpler version for clarity.





---

**Algorithm 1** Optimization algorithm for linear inverse problem.

---

Select $\epsilon$ and err$_\mathbf{x}$, then

$\mathbf{y} = \mathbf{y} - M\mathbf{x}^a$

$B = \text{err}_\mathbf{x} I_n$

$R = I_p$                                                    (possible extension to absolute/relative noise)

while $\frac{\sum \mathbf{x}_{neg}}{\sum \mathbf{x}_{pos}} < 0.0001$ and $iter < 50$ do

    $iter = iter + 1$

    solve minimization problem with cost function (7) with (8)

    $\widehat{\mathbf{x}} = \widehat{\mathbf{x}} + \mathbf{x}^a$

    for $j = 1 : n$

        if $\widehat{\mathbf{x}}_j \geq 0$

            $\mathbf{B}_{j,j} = \min\left\{\sqrt{1.5}\mathbf{B}_{j,j}, \text{err}_\mathbf{x}\right\}$

        else

            $\mathbf{B}_{j,j} = \mathbf{B}_{j,j}/2$

        end

    end

end

---

The estimation error caused by approximation (7) can be influenced by two choices of the user: (i) first guess $\mathbf{x}^a$, and (ii) regularization matrix $\Xi$. For a good choice of these values, the linear term should be close to zero (which would be the case for $\mathbf{x}^a$ approaching $\mathbf{x}$). The choice of parametric form of $\Xi$ corresponds to choosing a model of the SRS matrix error, since $\Xi = \Delta_\mathbf{M}^T \Delta_\mathbf{M}$.

In the following sections, we will discuss methods which estimate $\Xi$ from the data using parametrization of $\Xi$ by tridiagonal matrices with limited number of parameters. Specifically, we will investigate if the choice of $\mathbf{x}^a$ has an impact on better estimation of $\Xi$.

## 2.2 Optimization approach

In Eckhardt et al. (2008), the source term inversion problem is formulated as (7) with choices

$$\Xi = \mathbf{B} + \epsilon \mathbf{D}^T \mathbf{D}, \tag{8}$$

where matrix $\mathbf{B}$ is selected or estimated precision matrix, the matrix $\mathbf{D}$ a discrete representation of the second derivative with diagonal elements equal to $-2$ and equal to $1$ on the first sub-diagonals and the scalar $\epsilon$ is the parameter weighting the smoothness of the solution $\mathbf{x}$.

    Minimization of (7) does not guarantee the non-negativity of the estimated source term $\mathbf{x}$. To solve this issue, an iterative

procedure is adopted (Eckhardt et al., 2008) where minimization of (7) is done repetitively with reduced diagonal elements of $\mathbf{B}$ related to the negative parts of the solution, thus tightening the solution to the prior source term which is assumed to be





---

**Algorithm 2** LS-APC-VB algorithm for linear inverse problem.

1. Initialization:

   Set initial values (denoted by zero iteration number in superscript $^{(0)}$) of parameters used in the first iteration: $\langle \omega \rangle^{(0)} = \frac{1}{\max(M^T M)}$, $\langle \Upsilon \rangle^{(0)} = \mathrm{err}_{\mathbf{x}} I_n$, and $\langle L \rangle^{(0)} = I_n$.

2. Iterate from $i = 1$ until convergence or maximum number of iterations is reached:

   (a) Compute estimate of the source term $\langle \mathbf{x} \rangle^{(i)}$ using regularized least squares:

   $$\Sigma_{\mathbf{x}}^{(i)} = \left( \langle \omega \rangle^{(i-1)} M^T M + \left\langle L \Upsilon L^T \right\rangle^{(i-1)} \right)^{-1}, \tag{10}$$

   $$\mu_{\mathbf{x}}^{(i)} = \Sigma_{\mathbf{x}}^{(i)} \left( \langle \omega \rangle^{(i-1)} M^T \mathbf{y} + \left\langle L \Upsilon L^T \right\rangle^{(i-1)} \mathbf{x}^a \right), \tag{11}$$

   and respective moment of the truncated normal distribution.

   (b) Update estimates of $\langle \Upsilon \rangle^{(i)}$ and $\langle L \rangle^{(i)}$, using Eq. (A3)–(A10) defined in Appendix A.

   (c) Compute precision parameter $\langle \omega \rangle^{(i)}$ using Eq. (A11)–(A12) in Appendix A.

3. Report estimated source term $\langle \mathbf{x} \rangle$ and its uncertainty $\Sigma_{\mathbf{x}}$.

---

non-negative. The diagonal elements of $\mathbf{B}$ related to the positive parts of the solution can on the other hand be enlarged up to a selected constant. This can be iterated until the absolute value of the sum of negative source term elements is lower than $0.01\%$ of the sum of positive source term elements. Formally, $\mathbf{B}_{j,j}$ in the $i$the iteration is calculated as

$$\mathbf{B}_{j,j}^{(i)} = \begin{cases} 0.5 \mathbf{B}_{j,j}^{(i-1)} & \text{for } \mathbf{x}_j^{(i-1)} < 0, \\ \min\{\sqrt{1.5} \mathbf{B}_{j,j}^{(i-1)}, \mathrm{err}_{\mathbf{x}}\} & \text{for } \mathbf{x}_j^{(i-1)} \geq 0. \end{cases} \tag{9}$$

5   We observed very low sensitivity to the choice of the recommended values of $0.5$ and $\sqrt{1.5}$ in Eq. (9). In most cases, varying these values does not lead to any serious differences in the resulting estimate. However, the selection of parameters $\mathbf{x}^a$, $\mathrm{err}_{\mathbf{x}}$, and $\epsilon$ is crucial and it will be discussed in Sect. 2.4.

The method is summarized as Algorithm 1 and will be denoted as optimization method in this study. The maximum number of iterations is set to 50 which was enough for convergence in all our experiments. To solve the minimization problem (7), we
10   use the CVX toolbox (Grant and Boyd, 2008, 2018) for Matlab.

## 2.3 Bayesian approach

In Tichý et al. (2016), the problem was addressed using a Bayesian approach. The difference from the optimization approach is twofold. First, it has different approximation of the covariance matrix $\Xi$:

$$\Xi = \mathbf{L} \Upsilon \mathbf{L}^T, \tag{12}$$





where matrix $\mathbf{L}$ models smoothness and matrix $\Upsilon$ models closeness to the prior source term $\mathbf{x}^a$. Matrix $\Upsilon = \mathrm{diag}\,(v_1, \ldots, v_n)$ is a diagonal matrix with positive entries while matrix $\mathbf{L}$ is a lower bi-diagonal matrix

$$
\mathbf{L} = \begin{pmatrix} 1 & 0 & 0 & 0 \\ l_1 & 1 & 0 & 0 \\ 0 & \ddots & \ddots & 0 \\ 0 & 0 & l_{n-1} & 1 \end{pmatrix}.
\tag{13}
$$

Second, the Bayesian approach allows to estimate the hyper-parameters $\Upsilon$ and $\mathbf{L}$ from the data.

5    Specifically, it formulates a hierarchical probabilistic model:

$$
p(\mathbf{y}|\mathbf{x},\omega) = \mathcal{N}\left(\mathbf{Mx}, \omega^{-1}\mathbf{I}_p\right)
\tag{14}
$$

$$
p(\omega) = \mathcal{G}\left(\vartheta_0, \rho_0\right),
\tag{15}
$$

$$
p(\mathbf{x}|L,\Upsilon) = t\mathcal{N}\left(\mathbf{x}^a, \left(\mathbf{L}\Upsilon\mathbf{L}^T\right)^{-1}, [0, +\infty]\right),
\tag{16}
$$

$$
p(v_j) = \mathcal{G}\left(\alpha_0, \beta_0\right), \; j = 1, \ldots, n,
\tag{17}
$$

$$
p(l_j|\psi_j) = \mathcal{N}\left(-1, \psi_j^{-1}\right), \; j = 1, \ldots, n
\tag{18}
$$

$$
p(\psi_j) = \mathcal{G}\left(\zeta_0, \eta_0\right), \; j = 1, \ldots, n.
\tag{19}
$$

Here, prior constants $\alpha_0, \beta_0$ are selected similarly to $\vartheta_0, \rho_0$ as $10^{-10}$, yielding a non-informative prior, and prior constants $\zeta_0, \eta_0$ are selected as $10^{-2}$ to favor a smooth solution (equivalent to $l_j$ prior value $-1$), see discussion in (Tichý et al., 2016). To consider the prior vector $\mathbf{x}^a$ is novel in the LS-APC model.

15    To estimate the parameters of the prior model (14), (15), and (14)–(19), we will use two inference methods, variational Bayes approximation and Gibbs sampling.

### 2.3.1 Variational Bayes solution

Variational Bayes solution (Šmídl and Quinn, 2006) seeks the posterior in the specific form of conditional independence such as

$$
p(\mathbf{x}, \boldsymbol{v}, \mathbf{l}, \psi_{1,\ldots,n-1}, \omega|\mathbf{y}) \approx p(\mathbf{x}|\mathbf{y})p(\boldsymbol{v}|\mathbf{y})p(\mathbf{l}|\mathbf{y})
$$

$$
p(\psi_{1,\ldots,n-1}|\mathbf{y})p(\omega|\mathbf{y}). \tag{20}
$$



---

**Algorithm 3** LS-APC-G algorithm for linear inverse problem.

1. Initialization

    (a) Set an initial state for each variable $\mathbf{x}, \boldsymbol{\upsilon} \equiv \mathrm{err}_{\mathbf{x}}, \mathbf{l}, \psi_{1,\ldots,n-1}, \omega$.

    (b) Set iteration index $j = 1$.

2. Until the preselected number of iterations is reached

    (a) Sweep through all variables $\mathbf{x}, \boldsymbol{\upsilon}, \mathbf{l}, \psi_{1,\ldots,n-1}, \omega$ and sample from their respective full conditionals.

    (b) Update the parameters of full conditionals based on drawn samples.

3. Based on the histograms of samples, estimate the maximum, mean, median, variance or any other desired statistic.

---

The best possible approximation minimizes Kullback–Leibler divergence (Kullback and Leibler, 1951) between the estimated solution and hypothetical true posterior. This minimization uniquely determines the form of the posterior distribution

$$\tilde{p}(\mathbf{x}|\mathbf{y}) = t\mathcal{N}_{\mathbf{x}}\left(\mu_{\mathbf{x}}, \Sigma_{\mathbf{x}}\right), \tag{21}$$

$$\tilde{p}(\upsilon_j|\mathbf{y}) = \mathcal{G}_{\upsilon_j}\left(\alpha_j, \beta_j\right), \quad \forall j = 1,\ldots,n, \tag{22}$$

$$\tilde{p}(l_j|\mathbf{y}) = \mathcal{N}_{l_j}\left(\mu_{l_j}, \Sigma_{l_j}\right), \quad \forall j = 1,\ldots,n-1, \tag{23}$$

$$\tilde{p}(\psi_j|\mathbf{y}) = \mathcal{G}_{\psi_j}\left(\zeta_j, \eta_j\right), \quad \forall j = 1,\ldots,n-1, \tag{24}$$

$$\tilde{p}(\omega|\mathbf{y}) = \mathcal{G}_{\omega}\left(\vartheta, \rho\right), \tag{25}$$

where the shaping parameters $\mu_{\mathbf{x}}, \Sigma_{\mathbf{x}}, \alpha_j, \beta_j, \mu_{l_j}, \Sigma_{l_j}, \zeta_j, \eta_j, \vartheta, \rho$ are derived in Appendix A. The shaping parameters are functions of standard moments of posterior distribution which are here denoted as $\widehat{\mathbf{x}}$ and means expected value with respect to the distribution on the variable in the argument. The standard moments together with shaping parameters form a set of implicit equations which is solved iteratively, see Algorithm 2. Note, that only convergence to a local optimum is guaranteed, hence good initialization and iteration strategy is beneficial, see Algorithm 2 and discussion in (Tichý et al., 2016). The algorithm is denoted as LS-APC-VB algorithm.

### 2.3.2 Gibbs sampling solution

Gibbs sampler is a Monte Carlo Markov Chain method for obtaining sequences of samples from distributions from which direct sampling is difficult or intractable (Casella and George, 1992). It is a special case of the Metropolis-Hastings algorithm with proposal distribution derived directly from the model (Chib and Greenberg, 1995). Given a joint probability density $p(\mathbf{x}, \boldsymbol{\upsilon}, \mathbf{l}, \psi_{1,\ldots,n-1}, \omega, \mathbf{y})$, a full conditional distribution needs to be derived for each variable or a block of variables, i.e. for $\mathbf{x}$, distribution $p(\mathbf{x}|\boldsymbol{\upsilon}, \mathbf{l}, \psi_{1,\ldots,n-1}, \omega, \mathbf{y})$ has to be found. These full conditionals then serve as proposal generators and have the same form as (21)–(25). We use the original Gibbs sampler from (George and McCulloch, 1993). Having samples from the last iteration, or a random initialization for the first iteration, the algorithm sweeps through all variables and draws samples from





their respective full conditional distributions. It can be shown that samples generated in such a manner form a Markov chain whose stationary distribution, the distribution to which the chain converges, is the original joint probability density. Since the convergence of the algorithm can be very slow, it is common practice to discard the first few obtained samples. This is known as a burn-in period. The advantage of this algorithm is its indifference to the initial state from which sampling starts.

## 2.4 Tuning parameters and prior source term

All mentioned methods are sensitive to a certain extent to the selection of their parameters. Here, we will identify these tuning parameters and discuss their settings in the following experiments. Moreover, we will discuss the selection of the prior source term.

The optimization approach is summarized in Algorithm 1 where two key tuning parameters are needed, parameter $err_{\mathbf{x}}$ which affects the closeness of solution to the prior source term through the matrix $\mathbf{B}$ and parameter $\epsilon$ which affects the smoothness of solution. In the following experiments, we select the parameter $\epsilon$ by experience while it can be seen that the solution is similar for a relatively wide range of values (few orders of magnitude). The parameter $err_{\mathbf{x}}$ seems to be crucial for the optimization method and sensitivity to the choice of this parameter will be studied while $err_{\mathbf{x}}$ will be referred as the tuning parameter. Note that heuristic techniques such as the L-curve method (Hansen and O'Leary, 1993) can not be used here because of the modification of the matrix $\mathbf{B}$ within the algorithm. This will be demonstrated in Sect. 3, Fig. 2. The LS-APC-VB method, summarized in Algorithm 2, also needs the selection of initial $err_{\mathbf{x}}$, however, relatively low sensitivity to this choice was reported (Tichý et al., 2016). The LS-APC-G method, summarized in Algorithm 3, is also initialized using $err_{\mathbf{x}}$ while its sensitivity to this choice is negligible due to the Gibbs sampling mechanism.

To select the prior source term seems to be an even more difficult problem especially in cases of releases with limited available information. Therefore, we will investigate various errors in the prior source term which can be considered thanks to controlled experiments where the true source term is available. We consider time–shift of the prior source term in contrast with the true source term, different scale, and a blurred version of the true source term. These errors can be examined alone or combined which will probably be more realistic.

## 2.5 Tuning by cross-validation

While the tuning parameters selected in previous Section are often selected manually, statistical methods for their selection are also available. One of the most popular is cross-validation (**?**), which we will investigate in the context of source term determination. The methods is really simple, all available data are split in the training and testing data sets:, $\mathbf{y}_{\text{train}}, \mathbf{M}_{\text{train}}$, and $\mathbf{y}_{\text{test}}, \mathbf{M}_{\text{test}}$, respectively. The training dataset is then used for source term estimation while the test dataset is used for computation of the norm of the residue of the estimated source term, $||\mathbf{y}_{\text{test}} - \mathbf{M}_{\text{test}}\langle\mathbf{x}\rangle||_2$. Such estimate is known to be almost unbiased but with large variance. Therefore, the procedure is repeated several times and the tuning parameters are selected based on statistical evaluation of the results. In this experiment, we repeat random selection of 80% of measurements as the training set and using the remaining 20% as the test set. For each tuning parameter $err_{\mathbf{x}}$, this is repeated 100 times in order to reach statistical significance of the selected tuning parameter.

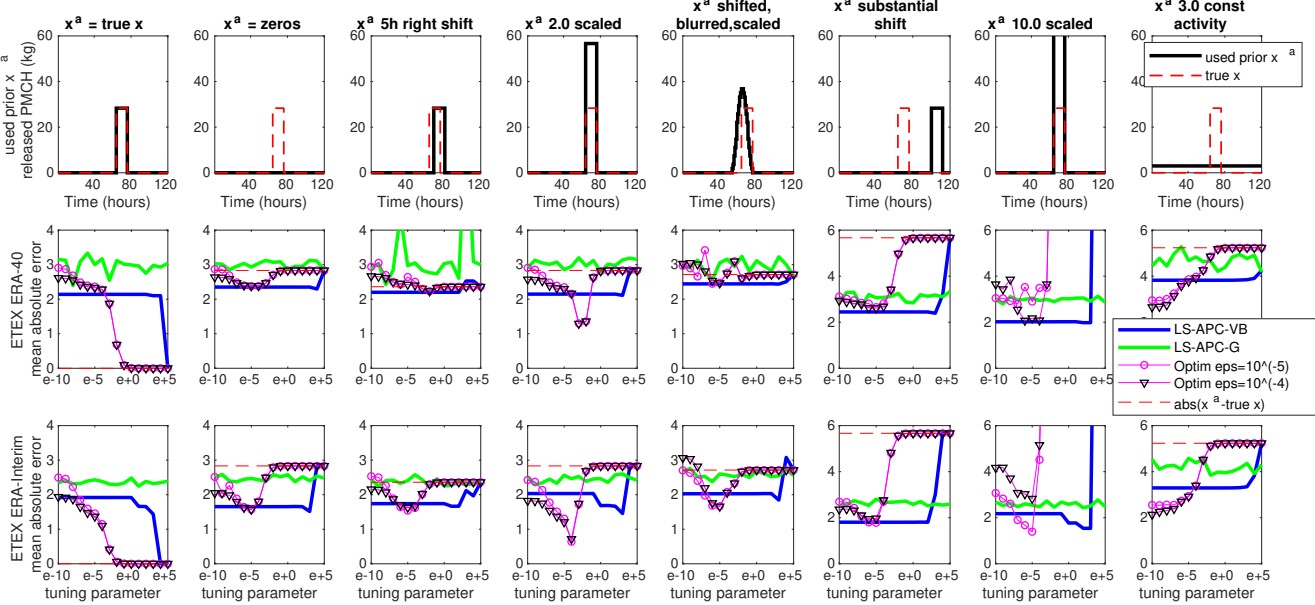

**Figure 1.** The uppermost row of panels shows eight different prior source terms $\mathbf{x}^a$ (black lines) used for the ETEX source term estimation. The true ETEX source term is repeated in every panel (red dashed line). The middle and the lowermost rows display mean absolute error between estimated and true source terms for ETEX ERA-40 and ETEX ERA-Interim datasets, respectively.

## 3   Sensitivity study on ETEX dataset

The European Tracer Experiment (ETEX) is one of a few large controlled tracer experiments (see https://rem.jrc.ec.europa.eu/etex/). We use data from the first release in which a total amount of 340 kg of nearly inert perfluoromethylcyclohexane (PMCH) was released at a constant rate for nearly 12 hours at Monterfil in Brittany, France, on 23 October 1994 (Nodop et al., 1998).

Atmospheric concentrations of PMCH were monitored at 168 measurement stations across Europe with a sampling interval of 3 hours and a total number of measurements 3102. The ETEX dataset has been used as a validation scenario for inverse modeling, see e.g. (Bocquet, 2007; Martinez-Camara et al., 2014; Tichý et al., 2016). The great benefit of this dataset is that the estimated source terms can be directly compared with the true release given in Fig. 1, first row, using dashed red lines.

To calculate the SRS matrices, we used the Lagrangian particle dispersion model FLEXPART (Stohl et al., 1998, 2005)

version 8.1. We assume that the release period occurred during 120 hours period, thus, 120 forward calculations of 1 hour hypothetical unit release were performed and SRS coefficients were calculated from simulated concentrations corresponding to the 3102 measurements. As a result, we obtained the SRS matrix $\mathbf{M} \in \mathbf{R}^{3102 \times 120}$. FLEXPART is driven by meteorological input data from the European Center for Medium-Range Weather Forecasts (ECMWF) where different datasets are available. We used two: (i) data from the 40-year re-analysis (ERA-40) and (ii) data from the continuously updated ERA-Interim re-

analysis.



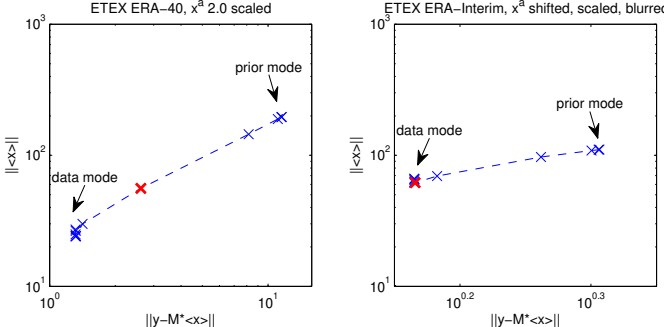

**Figure 2.** L-curve type plots using the optimization algorithm with $\epsilon = 1e - 5$ from ETEX ERA-40 with $\mathbf{x}^a$ 2.0 scaled (left) and from ETEX ERA-Interim with $\mathbf{x}^a$ shifted, scaled and blurred (right). The red crosses denote "sweet spots".

The tested method will be compared in terms of the mean absolute error (MAE) between the estimated and the true source term for different tuning parameters $\mathrm{err_x}$. We select a two representative values of the smoothing parameter $\epsilon$ for the optimization method. Specifically, we selected $\epsilon = 10^{-5}$ and $\epsilon = 10^{-4}$, while higher values lead to overly smooth, and lower values to non–smooth solutions. We tested 8 different prior source term $\mathbf{x}^a$, see Fig. 1, top row, black lines: $\mathbf{x}^a$ equal to (i) true source

term, (ii) zeros for all elements, (iii) true source term right-shifted by 5 time steps, (iv) true source term scaled by factor 2.0, (v) true source term blurred using convolution kernel of the size 5 times steps, left-shifted by 5 time-steps, and scaled by factor 1.3, (vi) true source term substantially right-shifted, (vii) true source term scaled by factor 10, and (viii) source term with constant activity. The resulting MAEs for all tested methods and for all 8 prior source terms are displayed in Fig. 1 for ETEX with ERA-40 dataset in the second column and for ETEX with ERA-Interim in the third row. The figures in the second and the

third rows are accompanied by the MAE between the true source term and the used prior source term displayed using dashed red lines.

### 3.1 Results

We observe that for all choices of the optimization method, the results exhibit two notable modes of solution: the data mode for tuning parameters with minimum impact on the loss function, and the prior mode for tuning parameter values that cause the

prior to dominate the loss function. This is notable for the results on the range of $\mathrm{err_x}$ in Fig. 1. For $\mathrm{err_x} = 10^{-10}$ the data term dominates the loss function, and all methods converge to a similar answer (note that the data mode is different for different smoothing parameters in the optimization method).

For $\mathrm{err_x} = 10^5$, the loss function is dominated by the prior and all estimates converge to $\mathbf{x}^a$. Although visibly stable modes of all the solutions are typically only two (data and prior mode), we also observe a third mode in the optimization solution, best

seen e.g. in Fig. 1 in the second row and the fourth column or in the third row and the fifth column, where the error significantly drops. These "sweet spots" are the desired locations that we hope to find by tuning of the hyper-parameters. While they are obvious when we know the ground truth, the challenge is to find them without this knowledge.





An attempt to find the optimal tuning via the L-curve method (i.e. dependence between the norm of the solution and the norm of the residuum between measurement and reconstruction) is displayed and demonstrated in two cases: ETEX ERA-40 with $\mathbf{x}^a$ 2.0 scaled (Fig. 2, left) and ETEX ERA-Interim with $\mathbf{x}^a$ shifted, scaled and blurred (Fig. 2, right) for optimization method with $\epsilon = 10^{-5}$. In these cases (and all others), L-curve shapes were not reached at all and thus an optimum regularization parameter

can not be chosen from these plots. The red crosses denote the value corresponding to minima of MAEs. One can see that the sweet spots are on the transition between the data mode and the prior mode of solutions with no specific feature in these measures. More detailed analysis is presented in the next Section.

The LS-APC-VB method also exhibits modes of solution; however, the transition between the data mode and the prior source term mode seems to be rather fast. Notably, no such transitions are observed in the case of the LS-APC-G method. This

is caused by the fact that the Gibbs sampling is not sensitive to the selection of the initial state, as discussed in Sect. 2.3.2. With the exception of $\mathbf{x}^a$ as a constant activity (Fig. 1, the eighth column), the LS-APC-VB method performs better than the optimization method when approaching the data mode of a solution. The LS-APC-G method suffers from overestimating the source term in time steps when the true source term is zero and not enough evidence is available in data. This can be clearly seen in Fig. 3 and Fig. 4 where estimates from the LS-APC-G method are displayed using green lines, see especially time steps

between 15 h and 40 h.

## 3.2   Desired optima of the the estimated source term

Here, we will discuss the behavior of the methods around the regions of the tuning parameter with minimum MAE (sweet spots) observed in the case of the optimization method. Note that no such regions are observed in cases of the LS-APC-VB and the LS-APC-G methods. The temporal profiles of the estimated source term at different penalization coefficients selected

around two different sweet spots are displayed in Fig. 3 and Fig. 4.

Fig. 3 displays results for the ETEX ERA-40 dataset with the prior source term selected as the 2.0 times scaled true source term. The top graph is a copy of sensitivity to tuning in terms of MAE from Fig. 1 (second row, fourth column), and labels (a), (b), (c), and (d) indicate selected values of tuning parameters for which the resulting estimated source terms are shown in Fig. 3 in the panels below the top one. The four estimates illustrate the transition from the data mode of solution (a), to the prior mode

of solution (d). The data mode underestimates the true release, while the prior mode overestimates it. As displayed in Fig. 3 (b) and (c), the slow transition between these two modes allows to approach the true source term closely, since the chosen prior term is only scaled version of the true release and the sweet spot lies exactly between the two modes. Both the LS-APC-VB and the LS-APC-G methods are diverging from the "good" solution since they consider it to be very unlikely with respect to the observed data. Since no heuristics such as the L-curve can identify this tuning as providing good results, see Fig. 2, left, we

we argue that choosing the optimal setting of the tuning parameter is not possible without knowledge of the true source term and the occurrence of the sweet spot is only a coincidence.

Fig. 4 displays results for the ETEX ERA-Interim dataset with the prior source term shifted, scaled, and blurred in the same way as in the third row and fifth column of Fig. 3. Here, the transition is not so sharp as in the previous case since the true source term does not lie exactly on the transition between the data mode, panel (a), and the prior mode, panel (d). The data



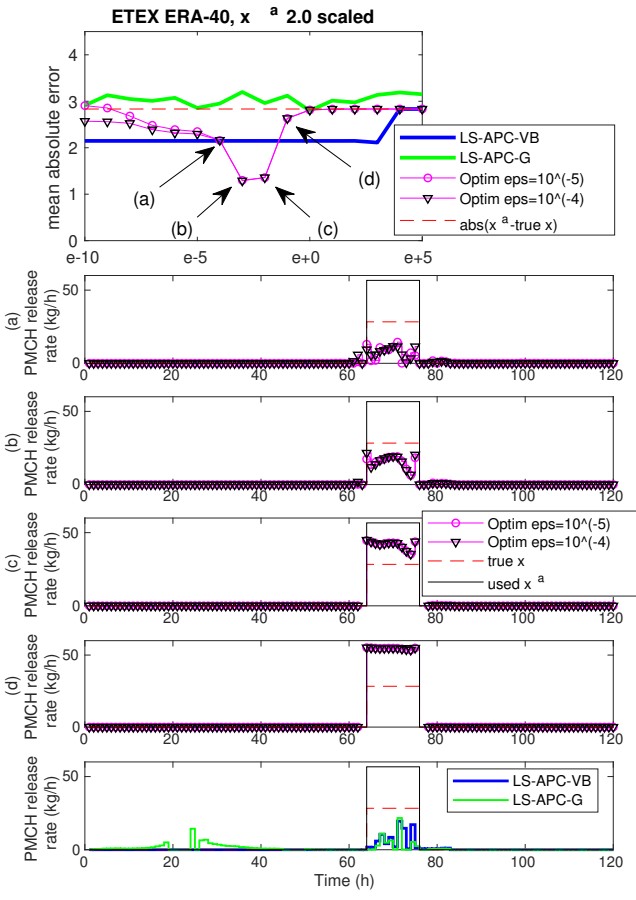

**Figure 3.** The uppermost panel shows mean absolute errors between estimated and true source terms for ETEX ERA-40 dataset with $\mathbf{x}^a$ 2.0 scaled for all methods. Certain settings of the tuning parameter are highlighted and labeled with (a), (b), (c) and (d). Estimated source terms for these tuning parameter choices are given in the panels below. The lowermost panel displays the estimated source terms from the LS-APC-VB and the LS-APC-G algorithms for comparison.

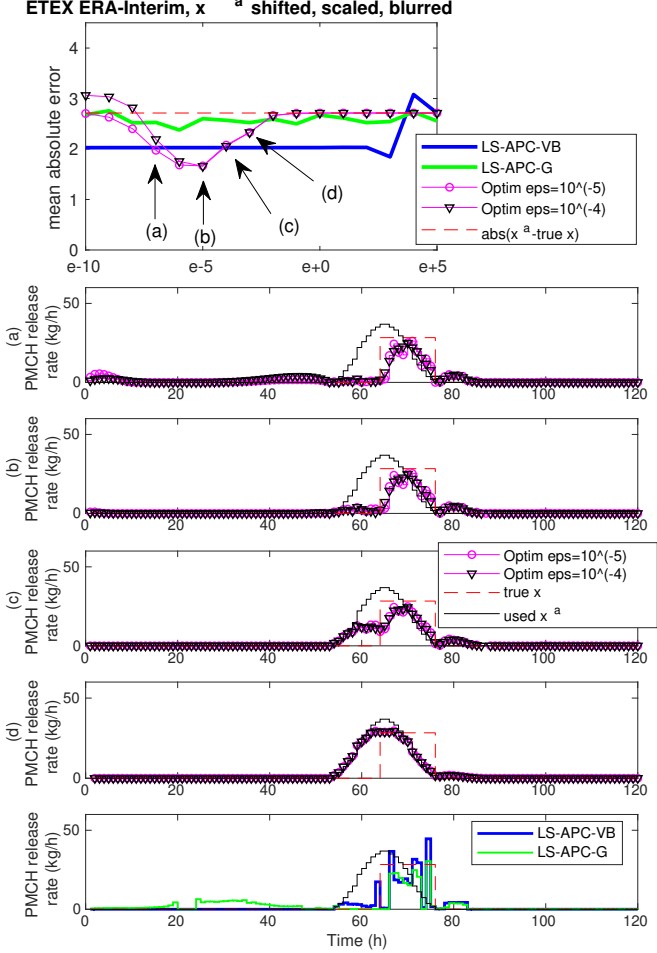

**Figure 4.** Same as Fig. 3 for ETEX ERA-Interim dataset with $\mathbf{x}^{\alpha}$ shifted, scaled and blurred.

mode (a) contains also non-zero elements mainly in the first half of the source term. The transition can be seen in Fig. 4 (b) and
(c) where the non-zero activity at the beginning of the data mode is eliminated by using prior source term information while
the non-zero elements are relatively close to the true release (b). In (c), the zero activity in the first half remains due to the prior
source term, however, the estimated activity within the true release period moves toward the assumed prior source term. In (d),
the estimation is already very close to the chosen prior source term. Once again, the improvement appears to be coincidental
rather than systematic.

We note that the two discussed sweet spots are selected as representative cases and other observed sweet spots (see e.g. Fig.
3, the second or the eighth columns) are very similar in nature. By analyzing the sweet spots, we conclude that they represent
a transition from the data mode to the prior mode of solution. In some cases, the transition is very close to the true release, see



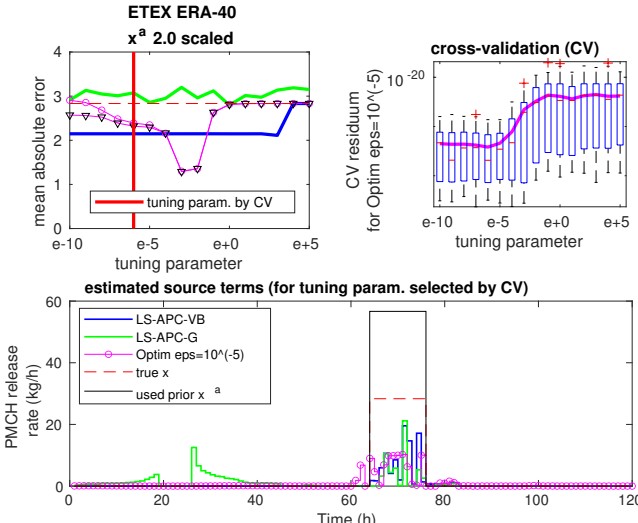

**Figure 5.** The top left panel shows the sensitivity of MAE to the tuning parameter for ETEX ERA-40 dataset with $\mathbf{x}^a$ 2.0 scaled. This is a repetition from Fig. 1. The chosen optimal setting based on CV is shown with a thick red vertical line. The top right panel shows the error residuals of the CV experiments as a function of the tuning parameter. Residuals are shown as box-and-whiskers plots, where the boxes extend between 25 and 75 percentiles (whiskers between 2.7 sigmas) and medians are marked with red lines while mean values are displayed using solid magenta line. The lowermost panel shows the source term obtained with the tuning parameter setting chosen via CV.

e.g. Fig. 3, while in some cases, no point on the transition path approaches the true solution, see e.g. Fig. 4, and the data or prior mode is the closest.

### 3.3 Tuning by cross-validation

Since the LS-APC-VB and the LS-APC-G methods provide rather stable estimates of the source term, we will investigate

the use of cross-validation (CV) on the optimization-based approaches. The results of CV for the optimization method with $\epsilon = 10^{-5}$ for selected combinations (i) ERA-40 with $\mathbf{x}^a$ 2.0 scaled, (ii) ERA-Interim with $\mathbf{x}^a$ shifted, blurred and scaled, and (iii) ERA-40 with $\mathbf{x}^a$ equal to the true source term are displayed in Fig. 5, 6, and 7 respectively at the top right panels. The results are displayed using boxplots where medians are displayed using red lines inside boxes while the boxes cover the 25th and the 75th quantiles. The mean values of the residuals for each tuning parameter are displayed using magenta lines. The

value of the tuning parameter that minimizes the CV error is also visualized in the top left panels using a solid vertical red lines inside the graphs of MAE sensitivity from Fig. 1. Bottom panels of figures display the estimated source terms using the tested methods for the tuning parameter selected by cross-validation together with the true source term (dashed red line) and the used prior source term (full black line).

     The results demonstrate significant differences between the prior mode and the data mode of the solution which can be seen

on all cross-validation boxplots. This is also the case $\mathbf{x}^a$ that are not displayed here. Notably, the minima of cross-validation





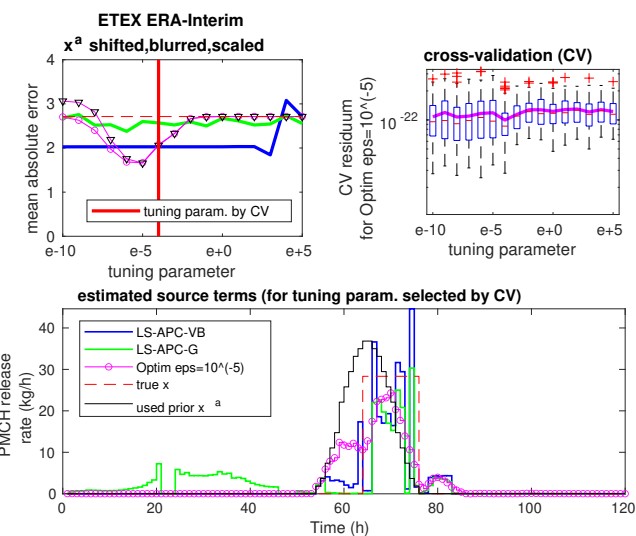

**Figure 6.** Same as Fig. 5 for ETEX ERA-Interim dataset with $\mathbf{x}^a$ shifted, scaled and blurred.

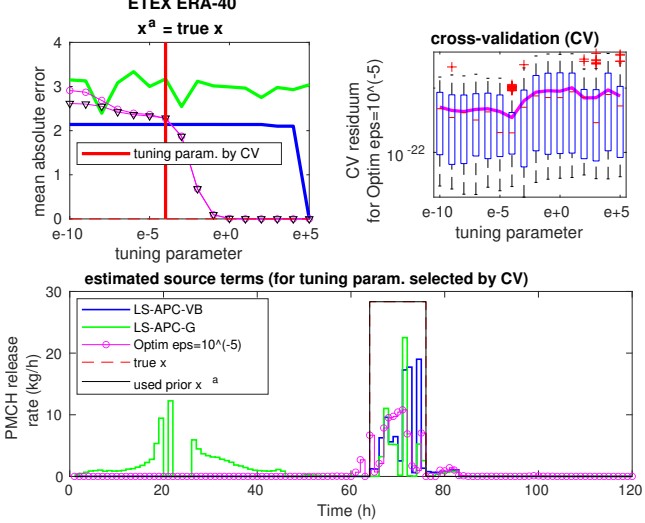

**Figure 7.** Same as Fig. 5 for ETEX ERA-40 dataset with $\mathbf{x}^a$ equal to true source term.





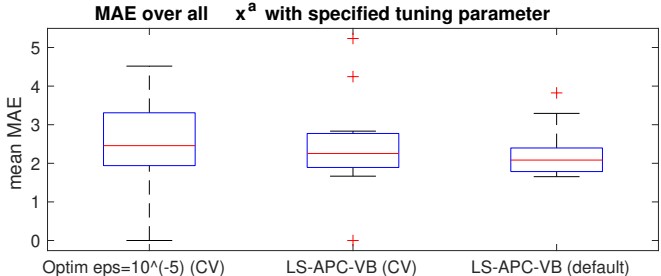

**Figure 8.** Box-and-whiskers plots of the MAE averaged over all explored prior source terms, with the tuning parameter settings determined by CV for the optimization method (left) and the LS-APC-VB method (middle), and for the LS-APC-VB method using default $err_x$ setting of $10^0$.

are not reached in positions of the sweet spots, indicating that the observed MAE minima are coincidental. In all tested cases, the minima of cross-validation are reached closer to the the data mode than to the prior mode. This is demonstrated on the extreme case of $\mathbf{x}^a$ equal to the true source term in Fig. 7. Even for this case, the minimum of cross-validation is associated with the data mode rather than the prior mode.

To evaluate the overall results, we compute the mean MAE over all estimated source terms using optimization method with $\epsilon = 10^{-5}$ with the tuning parameter $err_x$ selected using cross-validation (CV) for each prior source term $\mathbf{x}^a$. This result is given in Fig. 8 using box-and-whiskers plot. The same box-and-whiskers plots are also computed for the LS-APC-VB method with the same scheme of selection of tuning parameters $err_x$ using cross-validation method (denoted by (CV) in Fig. 8) and for the LS-APC-VB algorithm with the tuning parameter $err_x$ set to $10^0$ as recommended in (Tichý et al., 2016) (denoted by

(default) in Fig. 8). These results suggest the LS-APC-VB method with fixed start (but weighted to data using selection of $\omega^{(0)}$) slightly outperforms other methods in terms of the mean MAE on ETEX data with various assumed prior source terms without necessity of exhaustive tuning.

## 4   Sensitivity study on Chernobyl dataset

We demonstrate the sensitivity of the tuning methods on estimation of the source term of the Chernobyl accident which became,

together with the Fukushima accident, a widely discussed case in inverse modeling literature. Specifically, we study caesium-134 (Cs-134) and caesium-137 (Cs-137) releases from the Chernobyl nuclear power plant (ChNPP). For this purpose, we use a recently published dataset (Evangeliou et al., 2016) consisting of 4891 observations of Cs-134 and 12281 observations of Cs-137. We use the same experimental setup as in (Evangeliou et al., 2017) which will be now briefly described.

### 4.1   Atmospheric modeling

The Lagrangian particle dispersion model FLEXPART version 10.3 (Stohl et al., 1998, 2005; Pisso et al., 2019) was used to simulate the transport of radiocesium and to calculate the SRS matrices. FLEXPART was driven by two meteorological



reanalyses so that these can be compared. First, ERA-Interim (Dee et al., 2011), a European Center for Medium-range Weather Forecast (ECMWF) reanalyses, was used with temporal resolution of 3 hours and spatial resolution of circa 80 km on 60 vertical levels, from surface up to 0.1 hPa. Second, the ERA-40 (Uppala et al., 2005), ECMWF reanalysis, was used at a 125 km spatial resolution. The emissions from the ChNPP are discretized into six 0.5 km high vertical layers from 0 to 3 km. The

assumed temporal resolution is 3 hours from 0000 UTC on April 26 to 2100 UTC on May 5, 1986, for which the forward runs of FLEXPART are done. This period is selected according to the consensus that the activity declined by approximately six orders of magnitude after May 5 (De Cort et al., 1998). This discretized the temporal–spatial domain to 480 assumed releases (80 temporal elements times 6 vertical layers) for each nuclide. For each spatio-temporal element, concentrations and depositions sensitivities are computed using 300.000 particles. Following (Evangeliou et al., 2017), the aerosol tracers Cs-134

and Cs-137 are subject to wet (Grythe et al., 2017) and dry (Stohl et al., 2005) deposition depending on different particle sizes with aerodynamic mean diameters of 0.4, 1.2, 1.8, and 5.0 $\mu$m. The distribution of mass is assumed as 15%, 30%, 40%, and 15% for 0.4, 1.2, 1.8, and 5.0 $\mu$m particle sizes respectively, following measurements of (Malá et al., 2013) and computation results of (Tichý et al., 2018).

### 4.2   Prior source term and measurements uncertainties

The exact temporal profile of the Chernobyl release is uncertain although certain features are commonly accepted (De Cort et al., 1998). The first three days correspond to higher release with initial explosion and release of part of the fuel. The next four days correspond to lower releases and the last three days correspond to higher releases again due to fires and core meltdown. After these ten day, the released activity dropped by several orders of magnitude (De Cort et al., 1998).

We follow the setup of Evangeliou et al. (2017) and consider six previously estimated Chernobyl source terms of Cs-134

and Cs-137 where the criterion of consideration was their sufficient temporal resolution and emission height information. The source terms are taken from Brandt et al. (2002) (two cases with the same amount of release with slightly different release heights), Persson et al. (1987), Izrael et al. (1990), Abagyan et al. (1986), and Talerko (2005). See Evangeliou et al. (2017) for their detailed descriptions and profiles. The prior source term in our experiment is computed as their average emissions at a given time and height. In sum, the total prior releases of Cs-134 and Cs-137 are 54 PBq and 74 PBq, resprectively.

The uncertainties associated with measurements are relatively high since both concentration and deposition measurements are used from the dataset (Evangeliou et al., 2016). As was pointed out by Gudiksen et al. (1989); Winiarek et al. (2014), deposition measurements may be biased by unknown mass of radiocesium already deposited over Europe from e.g. nuclear weapons tests. This mass has been, however, reported (De Cort et al., 1998) and already subtracted from the dataset. Still, similarly to Evangeliou et al. (2017), we consider relative measurement errors of 30% for concentration measurements and

60% for deposition measurements while the absolute measurement errors are handled in the same way as in Stohl et al. (2012).

### 4.3   Results

In this case, direct comparison of the estimates with the true emission profile is not possible since this remains unknown. Therefore, we will provide results of the tested methods as sensitivity of the total estimated release activity to tuning parameters





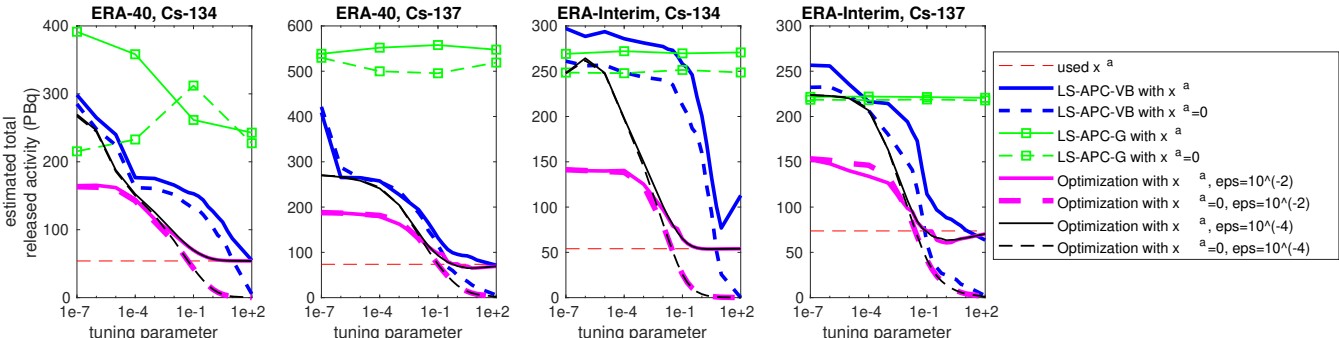

**Figure 9.** Estimated total released activities for both meteorological reanalyses ERA-40 and ERA-Interim and both nuclides Cs-134 and Cs-137 using all tested methods, see label bar on the right for line description.

in the same way as in Sect. 3. Note that the total release activity is a sum of releases from all six vertical layers and all four aerosol size fractions. Due to this composition of the problem, the selection of the smoothness parameter $\epsilon$ in the case of the optimization approach is relatively difficult since specific selection may fit better for one vertical layer than for another. We will provide results for two settings of this parameter, $\epsilon = 10^{-2}$ and $\epsilon = 10^{-4}$, leading to two different behavior of the optimization

method.

The resulting estimates of the total released activity is displayed in Fig. 9 where the total of the used prior source term $\mathbf{x}^a$ is displayed using dashed red line (same for all tested settings of the tuning parameter $\text{err}_\mathbf{x}$). The estimated total release activity with the use of the prior source term $\mathbf{x}^a$ is displayed using full lines with colors given in the legend in Fig. 9 while estimations without the use of this prior source term, i.e. with $\mathbf{x}^a = \mathbf{0}$, is given using dashed lines and respective colors.

Similarly to the ETEX results, the results in Fig. 9 suggest the occurrence of two main modes of solution, the data mode and the prior mode, with a smooth transition between them in the cases of LS-APC-VB and optimization methods. The LS-APC-G method (evaluated only at four points denoted by green squares due to high computational costs) has, again, low sensitivity to the initialization of the tuning parameter. However, the results of the LS-APC-G method are close to the data mode of the remaining methods, or higher than those. Contrary to the previous results, the LS-APC-VB algorithm does not provide a stable

solution and suffers from the need to select the tuning parameter. This signifies that the problem is ill conditioned even with the proposed regularization term, thus VB is converging to various local minima. The optimization method with both settings of the smoothness parameter has also two modes of solution. In the prior mode of solution (higher values of the tuning parameter), both settings approach the same total release activity for both non-zero (full lines) and zero (dashed lines) prior source terms. The prior mode is dominated by the used prior source term for an arbitrary smoothness parameter $\epsilon$. The difference can be

seen in the data mode where about one-third higher total released activity was estimated for smoothness parameter $\epsilon = 10^{-4}$ than for smoothness parameter $\epsilon = 10^{-2}$ on the same level of tuning parameter $\text{err}_\mathbf{x}$. This is caused by the penalization of high peaks of activity in the case of $\epsilon = 10^{-2}$. Thus, in the data mode of solution, the smoothness parameter is much more important than the used prior source term which plays almost no role here.



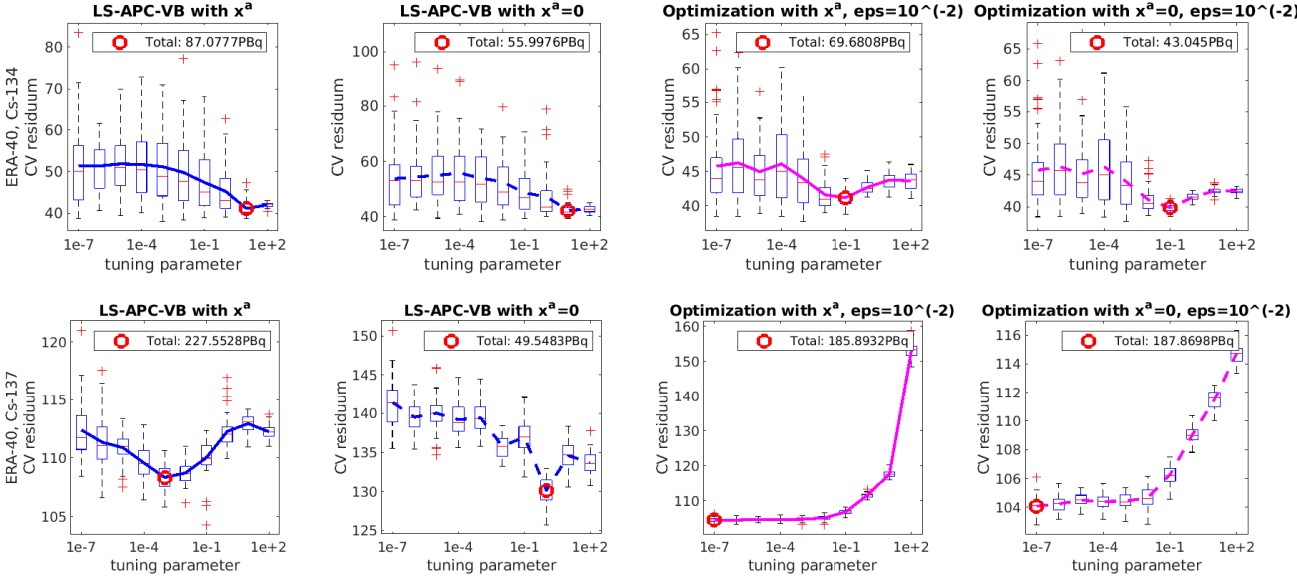

**Figure 10.** Cross-validation for Chernobyl Cs-134 (top panels) and Cs-137 (bottom panels) source terms using FLEXPART driven with ERA-40 meteorological reanalyses. Optima in the sense of cross-validation are denoted using red circles with total estimated releases reported in the legends.

Notice that the estimated mass is higher in the data mode than in the prior mode. This means that the model constrained by the measurement data alone would support a higher total release amount than the a priori source term. The true source term is not known, however, it is likely that the data mode overestimates the true total release. This can happen when the SRS matrix is biased. For instance, too rapid removal of radiocesium would lead to too low estimated air concentrations with

5 the correct source term, and the inversion would compensate the bias by increasing the posterior source term (notice, though, that deposition values would in this case be overestimated at least close to the source, leading to the contrary effect for the deposition data). Regardless, this effect shows that in the data mode, the resulting source term is heavily influenced by possible biases in the transport model.

### 4.4 Tuning by cross-validation

The same cross-validation scheme as in the case of the ETEX experiment, Sect. 3.3, was used here for the Chernobyl datasets. The train/test spilt was once again 80%/20% and the CV was performed 50 times for each tuning parameter $\text{err}_{\mathbf{x}}$. The cross-validation errors are displayed in Fig. 10 using box plots and associated mean values of the residue errors $||\mathbf{y}_{\text{test}} - \mathbf{M}_{\text{test}}\langle\mathbf{x}\rangle||_2$. Here, the results are given for the datasets of Cs-134 (top row) and Cs-137 (bottom row) with FLEXPART driven with ERA-40 meteorological fields. We will investigate CV on tuning of parameters for the optimization as well as the LS-APC-VB method.

The results are presented for two $\mathbf{x}^a$ configurations in Fig. 10, from the left: LS-APC-VB with $\mathbf{x}^a$, LS-APC-VB with $\mathbf{x}^a = \mathbf{0}$, the optimization method with $\mathbf{x}^a$ and with smoothness parameter $\epsilon = 10^{-2}$ and the optimization method with $\mathbf{x}^a = \mathbf{0}$ and with





smoothness parameter $\epsilon = 10^{-2}$. For these, boxplots are displayed together with mean residuals using the same types of lines as in Fig. 9. Moreover minimal mean residuals are identified and denoted using red circles in Fig. 10 for each graph and their associated total activities are displayed in the legend of each graph.

In the case of Cs-134 (top row), the cross-validation was able to determine optimal values of tuning parameters in cases of all tested methods. The total estimated releases associated with these tuning parameters are 87.1 PBq (LS-APC-VB with $\mathbf{x}^a$), 56 PBq (LS-APC-VB with $\mathbf{x}^a = \mathbf{0}$), 69.7 PBq (the optimization method with $\mathbf{x}^a$), and 43 PBq (the optimization method with $\mathbf{x}^a = \mathbf{0}$) which are in accordance with the mean of previously reported total activity 54 PBq used as a prior. Note that the prior dominated modes have lower residuals than the data dominated modes in all cases. This suggests that the used prior source term applied to the FLEXPART/ERA-40 simulation matches well with the measurements. On the other hand, this is not the case for Cs-137 where the prior dominated modes have, with the exception of LS-APC-VB with $\mathbf{x}^a = \mathbf{0}$, significantly higher residuals than the data dominated modes. This may be caused by two factors. First, the prior source term is less adequate for interpretation of measurements of Cs-137 than those of Cs-134. Second, all methods assume quadratic loss function which may be less appropriate for this dataset and caused overestimation of the source term with the tuning parameter selected using cross-validation in comparison with previously reported 74 PBq used as a prior. We note that similar results were also observed with the ERA-Interim dataset.

The results suggest that a well selected prior source term can bind the solution to acceptable values and prevents the occurrence of extreme outliers. On the other hand, we observed that the regularization terms commonly used to compensate errors of the SRS matrices are not able to compensate the error caused by inaccurate SRS matrices. Further research is clearly needed to develop more relevant method of regularization.

## 5 Conclusions

Methods for determination of source term of an atmospheric release have to cope with inaccurate prediction model, represented often by the source-receptor sensitivity (SRS) matrix. Relying solely on the SRS matrix using best estimate of weather and dispersion parameters may lead to highly inaccurate results. We have shown that various regularization terms introduced by different inversion methods are essentially coarse approximation of the error of the SRS matrix and thus we can evaluate their suitability using methods of statistical model validation. We have performed sensitivity of inverse modeling methods to the selection of the prior source term (first guess), and other tuning parameters for two selected inversion methods, the optimization method (Eckhardt et al., 2008), and the LS-APC method (Tichý et al., 2016), on datasets from the ETEX controlled release, and the Chernobyl releases of caesium-134 and caesium-137.

We have observed that the results have two strong modes of solution, the data mode for minimal influence of prior on the loss, and the prior mode for loss function with significant influence of the prior. The prior mode is naturally significantly influenced by the choice of the prior source term. However, the dominant impact on the resulting estimate has the choice of the regularization. In the case of the ETEX dataset, good estimates were obtained for every choice of the prior source term, however, the regularization has to be carefully tuned. For some choices of the prior source term, the error of the estimated



source term was exceptionally low for good selection of the tuning parameters. After analyzing these minima, we conjecture that they are caused by coincidence. These minima are visible only in comparison with the ground truth, they have no visible impact on the common validation metrics such as the L-curve or cross-validation and thus cannot be identified objectively.

We have tested suitability of the cross-validation approach for selection of the tuning parameters for both tested methods. In the case of the ETEX release, we have observed that this approach tends to select modes closer to the data mode than the prior mode of solution. However, this is not the case of the Chernobyl Cs-134 release where cross-validation selects solutions close to the prior dominated mode. This may be caused by the fact that the used prior source term here well fits the measurements and only small corrections by the inversion are needed.

An interesting question is whether it is beneficial to use a non-zero prior source term at all. Considering the ETEX experiment, where the true release is known, one can see that the estimates in data modes are often even better than the considered prior source terms. On the other hand, when the used prior source term is close to the true release, which is probably the case for the Chernobyl Cs-134 release, its use seems beneficial. Also, the prior source term could be valuable in cases when the release is not fully seen by the measurement network and thus the measurements do not provide a good constraint for the source term estimation. However, to determine the reliability of the prior source term is difficult and even impossible in real world scenarios and the prior source term would be probably shifted, scaled, and/or blurred. This task can be tackled using cross-validation approach providing a reasonable although computationally expensive tool for determination at least between a prior dominated mode or a data dominated mode of solution.

*Code and data availability.* All data used for the present publication can be freely downloaded from https://rem.jrc.ec.europa.eu/etex/ and from the supplement of Evangeliou et al. (2016), respectivelly. The FLEXPART models version 8.1 and version 10.3 are open source and freely available from their developers from https://www.flexpart.eu/. Reference MATLAB implementations of algorithms can be downloaded from http://www.utia.cas.cz/linear_inversion_methods/.





## Appendix A: Shaping parameters of LS-APC-VB posteriors

$$\Sigma_{\mathbf{x}} = \left( \langle \omega \rangle M^T M + \langle L \Upsilon L^T \rangle \right)^{-1}, \tag{A1}$$

$$\mu_{\mathbf{x}} = \Sigma_{\mathbf{x}} \left( \langle \omega \rangle M^T \mathbf{y} + \langle L \Upsilon L^T \rangle \mathbf{x}^a \right), \tag{A2}$$

$$\alpha = \alpha_0 + \frac{1}{2} \mathbf{1}_{n,1}, \tag{A3}$$

$$\beta = \beta_0 + \frac{1}{2} \mathrm{diag} \left( \langle L^T \mathbf{x} \mathbf{x}^T L \rangle \right) \tag{A4}$$

$$- \mathrm{diag} \left( \langle L^T \mathbf{x}^a \mathbf{x}^T L \rangle \right) + \frac{1}{2} \mathrm{diag} \left( \langle L^T \mathbf{x}^a \mathbf{x}^{aT} L \rangle \right), \tag{A5}$$

$$\Sigma_{l_j} = \left( \langle \upsilon_j \rangle \langle x_{j+1}^2 \rangle + \langle \psi_j \rangle \right)^{-1}, \tag{A6}$$

$$\mu_{l_j} = \Sigma_{l_j} \Bigg( - \langle \upsilon_j \rangle \langle x_j x_{j+1} \rangle + (-1) \langle \psi_j \rangle \tag{A7}$$

$$+ \langle \Upsilon \rangle_{j,j} \left( \mathbf{x}_j^a \langle \mathbf{x} \rangle_{j+1} + \mathbf{x}_{j+1}^a \langle \mathbf{x} \rangle_j - \mathbf{x}_j^a \mathbf{x}_j^a \right) \Bigg), \tag{A8}$$

$$\zeta_j = \zeta_0 + \frac{1}{2}, \tag{A9}$$

$$\eta_j = \eta_0 + \frac{1}{2} \langle (l_j - l_0)^2 \rangle, \tag{A10}$$

$$\vartheta = \vartheta_0 + \frac{p}{2}, \tag{A11}$$

$$\rho = \rho_0 + \frac{1}{2} \mathrm{tr} \left( \langle \mathbf{x} \mathbf{x}^T \rangle M^T M \right) - \frac{1}{2} 2 \mathbf{y}^T M \langle \mathbf{x} \rangle + \frac{1}{2} \mathbf{y}^T \mathbf{y}. \tag{A12}$$

*Author contributions.* OT designed and performed the experiments and wrote the paper. LU performed Gibbs sampling experiments and wrote parts of the paper. VŠ designed and supervised the study and wrote parts of the paper. NE prepare Chernobyl dataset and commented on the manuscript. AS commented on the manuscript and wrote parts of the paper.

*Competing interests.* The authors declare that they have no conflict of interest.

*Acknowledgements.* This work was supported by the Czech Science Foundation, grant no. GA20-27939S.





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
