# Peer review of "On tuning of atmospheric inverse methods: Comparisons on ETEX and Chernobyl datasets using the atmospheric transport model FLEXPART"

_Geoscientific Model Development, 2020_

## Referee Comment (RC1) · Anonymous Referee #1 · 23 Aug 2020

This paper presents sensitivity studies of source term estimations applied to ETEX experiment and Chernobyl accident. Interesting results are shown and they shed light on the importance of the regularization parameters. While some well know aspects of the inverse problems are discussed in the paper, it is not clear what is suggested for the future applications. For the ETEX experiment, the source term estimation results are pretty disappointing. More discussions are probably needed for the causes.

General:

The derivation of the equations (4-6) is important to reach the final simplified form (7). However, only the simplified form is used and discussed later in the manuscript. It may
be more appropriate to put them in an appendix.

Most of the estimated source terms are not as good. It is likely due to the model errors in the SRS matrix M. It will be useful to present the M matrix or its simplified form for both applications.

Two versions of the FLEXPART are used for the two different applications. How different are the two versions? If version 10.3 is considered an updated version, will replacing v8.1 with v10.3 for ETEX experiment provide better results?

Specific and editorial:

Page 2, lines 18-20: Interpreting no prior source term as having zero prior source term is not accurate.

Page 4, Equation 5: Some terms are probably missing. Is it "$\bar{x}\text{-}x_a$" or "$\bar{x}_+x_a$"?

Page 4, Equation 6: Please check whether it is correct.

Page 4, Equation 7: J is supposed to be the sum of the two terms, rather than the difference.

Page 7, Equations 14-19: The notations, such as the normal/Gaussian, truncated normal, and Gamma distributions, should be explained.

Page 9, line 26: Please fix "?".

Page 11, line 2: Remove "a" before "two representative values".

Page 12, line 30: Remove the repeated "we".

Page 18, line 30: Please briefly explain how the absolute measurement errors are handled.
* * *

---

## Referee Comment (RC2) · Anonymous Referee #2 · 28 Aug 2020

The paper "On tuning of atmospheric inverse methods: Comparison on ETEX and Chernobyl datasets using FLEXPART v8.1 and v10.3" addresses a very important aspect of emergency management, the reconstruction of the mostly unknown source term. It discusses in detail the importance of prior knowledge which is often very small in real situation.

The paper discusses tuning approaches of some of the key parameters. This is discussed in the main part of the paper, however, more from a theoretical perspective. The paper would gain from a discussion on the application for a novel source term that has no a priory knowledge. Which steps should be performed to optimise the parameters

for the reconstruction of the source terms?

Otherwise, the paper is very informative and does not require modification in structure and style.

The authors may also briefly discuss the difference between version 8.1 and 10.3 and to which extend the same functionality is available in the newest version 10.4

———————————————

---

## Author Comment (AC1) · 1 Oct 2020

We would like to thank you for providing us a detailed review of our manuscript. We are glad that we can submit a revision of our paper. In the following text, we will respond to all comments.

**This paper presents sensitivity studies of source term estimations applied to ETEX experiment and Chernobyl accident. Interesting results are shown and they shed light on the importance of the regularization parameters. While some well know aspects of the inverse problems are discussed in the paper, it is not clear what is suggested for the future applications. For the ETEX experiment,**

[Figure]

the source term estimation results are pretty disappointing. More discussions
are probably needed for the causes.

→ **The derivation of the equations (4-6) is important to reach the final simplified
form (7). However, only the simplified form is used and discussed later in the
manuscript. It may be more appropriate to put them in an appendix.**

*Authors response*: We agree that the derivation may be put into the appendix. We also
rewrite the residual term completely to the substituted variables $\overline{y}$ and $\overline{x}$ for bet-
ter clarity and understanding. The goal is to show that norms of residuum and
quadratic terms have the same forms in original and substituted variables, how-
ever, with different linear terms (which are typically ignored in optimization in both
cases).

*Changes made in the paper*: The derivation was reformulated and its core was added to
the appendix.

→ **Most of the estimated source terms are not as good. It is likely due to the
model errors in the SRS matrix M. It will be useful to present the M matrix or its
simplified form for both applications.**

*Authors response*: We completely agree with the reviewer that the estimated source
terms, especially in the case of ETEX release, is imperfect. However, we have
also shown that they can not be improved without a lucky coincidence by hitting
the right regularization setting. It is also a consequence of ill conditioned nature
of the problem. We agree that displaying singular values of the SRS matrix
illustrates the sensitivity to regularization of the inversion problem. We have also
tested the ETEX case using another atmospheric transport model, HYSPLIT.
Since the results were very similar to those using FLEXPART, we decided to not

used HYSPLIT data in the paper, however, we display here scatter plots between measurement and model reconstruction for both FLEXPART simulations and for HYSPLIT simulation, see Fig. 1. We conjecture that the main source of error is inaccurate reconstruction of the weather in 1994.

Similar information could be also provided for Chernobyl dataset, however, details of modeling and validation was already provided in (Evangeliou et al., 2017).

*Changes made in the paper*: We added a section to appendix with SRS matrices for the ETEX experiment.

→ **Two versions of the FLEXPART are used for the two different applications. How different are the two versions? If version 10.3 is considered an updated version, will replacing v8.1 with v10.3 for ETEX experiment provide better results?**

*Authors response*: Recent updates added new functionality in FLEXPART such as the possibility to run the model for skewed convective conditions, to run the model backward from deposition values, or parallelization. However, for the applications presented here, no important changes were made between v8.1 and v10.3, such that results of both versions can be expected to be equivalent.

The main focus of the paper is analysis of sensitivity of inversion methods to choices such as the prior source term or tuning parameters. We have not intended to compare versions of the FLEXPART. We realize that the previous title of the paper could be confusing since it mentioned both FLEXPART versions used in the paper and it might be understand as the comparison between FLEXPART versions. Therefore, we changed the title slightly after consultation with the editor.

[Figure]

*Changes made in the paper***:** We change the title of the paper slightly to avoid the misunderstanding about the comparison between FLEXPART versions. Now it reads "On tuning of atmospheric inverse methods: Comparisons on ETEX and Chernobyl datasets using the atmospheric transport model FLEXPART".

**Specific and editorial:**

→ **Page 2, lines 18-20: Interpreting no prior source term as having zero prior source term is not accurate.**

*Authors response***:** We agree that this interpretation was inappropriate and we rephrase the sentence.

→ **Page 4, Equation 5: Some terms are probably missing. Is it $\bar{x} - x_a$ or $\bar{x} + x_a$?**

*Authors response***:** There are no missing terms but there should be the plus sign instead of minus sign, thank you for noting that.

→ **Page 4, Equation 6: Please check whether it is correct.**

*Authors response***:** Same as previous comment, yes, there should be sign changed.

→ **Page 4, Equation 7: J is supposed to be the sum of the two terms, rather than the difference.**

*Authors response***:** Indeed, thank you for noting this typo.

→ **Page 7, Equations 14-19: The notations, such as the normal/Gaussian, truncated normal, and Gamma distributions, should be explained.**

*Authors response*: We agree that explanations of these symbols were missing and we added them to the text.

→ **Page 9, line 26: Please fix "?".**

*Authors response*: Thank you for noting this missing citation, we corrected it.

→ **Page 11, line 2: Remove "a" before "two representative values".**

*Authors response*: Removed.

→ **Page 12, line 30: Remove the repeated "we".**

*Authors response*: Removed.

→ **Page 18, line 30: Please briefly explain how the absolute measurement errors are handled.**

*Authors response*: The information is added to the manuscript.

———————————————

[Figure]

**Fig. 1.** Scatter plots between measurement and model reconstruction for both FLEXPART simulations and for HYSPLIT simulation.

---

## Author Comment (AC2) · 1 Oct 2020

We would like to thank you for providing us with review of our manuscript. We are glad that we can submit a revision of our paper. In the following text, we will respond to all comments.

**The paper "On tuning of atmospheric inverse methods: Comparison on ETEX and Chernobyl datasets using FLEXPART v8.1 and v10.3" addresses a very important aspect of emergency management, the reconstruction of the mostly unknown source term. It discusses in detail the importance of prior knowledge which is often very small in real situation.**

**→ The paper discusses tuning approaches of some of the key parameters. This is discussed in the main part of the paper, however, more from a theoretical perspective. The paper would gain from a discussion on the application for a novel source term that has no a priory knowledge. Which steps should be performed to optimise the parameters for the reconstruction of the source terms?**

*Authors response*: Existing methods of source term estimation are based on omitting linear terms in optimization criterion $J$ (considering the model of deviation of the SRS matrix as $\mathbf{M}_{true} = (\mathbf{M} + \Delta_{\mathbf{M}})$). Since the knowledge on $\Delta_{\mathbf{M}}$ is always insufficient, it seems to be a reasonable option. The regularization terms has to be used to improve conditioning of the linear inverse problem. Tuning the penalizations is often done manually without reporting alternative solutions. One of our main recommendations is to perform cross-validation to determine agreement of the chosen regularization with the data.

Second recommendation is for future development, in which we propose to use additional information, such as sensitivity of the concentration field around the sensors, to design new regularization terms that would to compensate the deviation $\Delta_{\mathbf{M}}$.

*Changes made in the paper*: We extended discussion in the Conclusion section accordingly.

**Otherwise, the paper is very informative and does not require modification in structure and style.**

**→ The authors may also briefly discuss the difference between version 8.1 and 10.3 and to which extend the same functionality is available in the newest version 10.4**
*Authors response*: Recent updates added new functionality in FLEXPART such as the possibility to run the model for skewed convective conditions, to run the model backward from deposition values, or parallelization. However, for the applications presented here, no important changes were made between v8.1 and v10.3, such that results of both versions can be expected to be equivalent.

The main focus of the paper is analysis of sensitivity of inversion methods to choices such as the prior source term or tuning parameters. We have not intended to compare versions of the FLEXPART. We realize that the previous title of the paper could be confusing since it mentioned both FLEXPART versions used in the paper and it might be understand as the comparison between FLEXPART versions. Therefore, we changed the title slightly after consultation with the editor.

*Changes made in the paper*: We change the title of the paper slightly to avoid the misunderstanding about the comparison between FLEXPART versions. Now it reads "On tuning of atmospheric inverse methods: Comparisons on ETEX and Chernobyl datasets using the atmospheric transport model FLEXPART".
* * *